# Supplementation with Red Wine Extract Increases Insulin Sensitivity and Peripheral Blood Mononuclear Sirt1 Expression in Nondiabetic Humans

**DOI:** 10.3390/nu12103108

**Published:** 2020-10-12

**Authors:** Munehiro Kitada, Yoshio Ogura, Itaru Monno, Daisuke Koya

**Affiliations:** 1Department of Diabetology and Endocrinology, Kanazawa Medical University, Daigaku, Uchinada, Ishikawa 920-0293, Japan; namu1192@kanazawa-med.ac.jp (Y.O.); imonno@kanazawa-med.ac.jp (I.M.); 2Division of Anticipatory Molecular Food Science and Technology, Medical Research Institute, Kanazawa Medical University, Daigaku, Uchinada, Ishikawa 920-0293, Japan

**Keywords:** red wine extract, resveratrol, polyphenols, insulin sensitivity, Sirt1

## Abstract

The aim of this study was to investigate the effects of dietary supplementation with a nonalcoholic red wine extract (RWE), including resveratrol and polyphenols, on insulin sensitivity and Sirt1 expression in nondiabetic humans. The present study was a single-arm, open-label and prospective study. Twelve subjects received supplementation with RWE, including 19.2 mg resveratrol and 136 mg polyphenols, daily for 8 weeks. After 8 weeks, metabolic parameters, including glucose/lipid metabolism and inflammatory markers, were evaluated. mRNA expression of Sirt1 was evaluated in isolated peripheral blood mononuclear cells (PBMNCs). Additionally, Sirt1 and phosphorylated AMP-activated kinase (p-AMPK) expression were evaluated in cultured human monocytes (THP-1 cells). Supplementation with RWE for 8 weeks decreased the homeostasis model assessment for insulin resistance (HOMA-IR), which indicates an increase in insulin sensitivity. Serum low-density lipoprotein-cholesterol (LDL-C), triglyceride (TG) and interleukin-6 (IL-6) were significantly decreased by RWE supplementation for 8 weeks. Additionally, Sirt1 mRNA expression in isolated PBMNCs was significantly increased after 8 weeks of RWE supplementation. Moreover, the rate of increase in Sirt1 expression was positively correlated with the rate of change in HOMA-IR. The administration of RWE increased Sirt1 and p-AMPK expression in cultured THP-1 cells. Supplementation with RWE improved metabolism, such as insulin sensitivity, lipid profile and inflammation, in humans. Additionally, RWE supplementation induced an increase in Sirt1 expression in PBMNCs, which may be associated with an improvement in insulin sensitivity.

## 1. Introduction

Metabolic derangement, including type 2 diabetes mellitus (T2DM), hypertension and dyslipidemia, which is based on insulin resistance, is closely related to the initiation and progression of cardiovascular disease (CVD) [1]. Therefore, maintaining metabolic health, including improving insulin sensitivity, is important to protect vascular tissues against metabolic-derangement-related cellular damage. Individual lifestyles, including dietary habits, affect metabolic and cardiovascular health. Appropriate consumption of red wine, 20–30 g/day as amount of alcohol, is thought to be part of a healthy lifestyle [2,3,4]. Previous epidemiological studies have shown an inverse association between dietary polyphenol consumption and mortality from CVD [5,6,7,8]. The components of red wine contain many polyphenols, which are a complex mixture of flavonoids such as anthocyanins and flavan-3-ols and nonflavonoids such as resveratrol, cinnamates and gallic acid [9]. Red wine polyphenols possess vasoprotective effects through anti-aggregatory platelet activity, antioxidant and anti-inflammatory properties, the generation and release of nitric oxide (NO) and glucose/lipid-metabolism-improving effects, which contribute to maintaining metabolic and cardiovascular health [10,11,12].

Aging is closely associated with metabolic derangement, including insulin resistance. Caloric restriction (CR)/dietary restriction (DR) retards aging or extends life spans [13]. The benefits of CR/DR for the suppression of age-related disorders, including glucose intolerance and CVD, have also been reported in rhesus monkeys and humans by improving insulin sensitivity and oxidative stress/inflammation [14,15,16,17]. Therefore, CR/DR mimetics may be anti-aging therapies, resulting in the maintenance of cardiometabolic health. Sirt1, a nicotinamide adenine dinucleotide (NAD^+^)-dependent deacetylase, has been identified as one of the possible molecules through which CR/DR exerts anti-aging effects [18,19]. Resveratrol, a polyphenolic phytoalexin that occurs in red wine, has been one of the most extensively studied Sirt1 activators, as one of the CR/DR mimetics [20] and is a critical constituent that contributes to the health benefits of red wine. Thus, polyphenols, including resveratrol from red wine, may be candidates to improve cardiometabolic alterations associated with aging due to their pleiotropic properties. However, there are few reports on whether red wine polyphenols, exert beneficial effects on glucose/lipid metabolism and Sirt1 activation in humans. In this study, we investigated the effects of red wine polyphenols on glucose/lipid metabolism and Sirt1 expression in isolated peripheral blood mononuclear cells (PBMNCs) using red wine extract (RWE).

## 2. Materials and Methods

### 2.1. Composition of RWE

The alcohol-free RWE was obtained from NATURE Supplement (Osaka, Japan). This RWE is derived from red wine produced in the Rhone valley regions of southern France. The polyphenol contents in the RWE were assessed by high-performance liquid chromatography (HPLC) analysis (Appendix A) and revealed that 166 mg of wine solids contained 9.6 mg of resveratrol and 68 mg of polyphenols: catechin 1.16 mg, epicatechin 0.83 mg, tannin 29.3 mg, quercetin glycoside 0.33 mg, malvidin glycoside 1.99 mg, total anthocyanin 5.15 mg, anthocyanin monomer 4.15 mg, anthocyanin polymer 1.00 mg per 1 capsule.

### 2.2. Subjects and Study Design

Participants were recruited through advertisements on local posters. Males or females who were 20–70 years old were eligible. The exclusion criteria included diabetes (HbA1c ≥ 6.5%); pre-existing endocrine, kidney, liver, heart and malignant disease; anemia (male: hemoglobin (Hb) < 10.0 g/dL, female: Hb < 9.0 g/dL); alcohol abuse; smoking; the use of medicines/supplements; and planned lifestyle changes. We enrolled 12 participants, including 8 males and 4 females, in this study.

This study is a single-arm, open-label, prospective study and conducted at Kanazawa Medical University Hospital. Subjects were treated for 8 weeks with 2 capsules of RWE (containing 9.6 mg resveratrol and 68 mg polyphenols per capsule) twice daily for a total of 19.2 mg resveratrol and 136 mg polyphenols per day. During the study period, participants were instructed to abstain from supplements and foods suspected to contain polyphenols in significant amounts and the adherence for them was confirmed every visit. Moreover, the importance of maintaining their normal way of life was underscored. Compliance, defined as the proportion of capsules ingested relative to the intended number, was calculated when participants returned the remaining capsules during the final visit.

### 2.3. Overall Visits and Interventions

Examinations were performed at baseline and 4 and 8 weeks after supplementation with RWE, with the same equipment and by the same physicians. When completing the physical examination, including routine clinical biochemistry data at baseline, capsules were provided and participants were instructed to initiate capsule consumption from the evening and twice daily. At week 4, potential adverse events were recorded and fasting blood samples were taken for safety purposes. In addition, participants visited the hospital on the examination day in the morning after overnight fasting at week 8 and then blood samples were collected.

### 2.4. Ethical Approval

Participants were given detailed explanations of the study protocol. Informed consent was obtained from each participant. The study protocol was approved by the Regional Committee on Health Research Ethics and the Ethical Committee of Kanazawa Medical University (IRB No. M229, Uchinada, Ishikawa, Japan) and conformed to the ethical principles set forth in the Declaration of Helsinki.

### 2.5. General Measurements

Body weight (BW) and body composition were measured using In Body^®^ (Biospace Japan, Inc., Tokyo, Japan) with the participants being lightly clothed; the participants urinated during the 30 min prior to the In Body^®^ assessment [21]. In addition, blood pressure (BP) and heart rate (HR) were measured in a sitting position after resting for 5 min [21]. Routine biochemistry and physical examinations were performed at screening to investigate the presence of exclusion criteria.

### 2.6. Blood Sample Analysis

Routine biochemistry (creatinine (Cr), uric acid (UA), aspartate aminotransferase (AST), alanine transaminase (ALT) and γ-glutamyl transpeptidase (γ-GTP)) parameters were analyzed continuously throughout the study at the Department of Clinical Biochemistry of Kanazawa Medical University Hospital using standard methods [21]. HbA1c and glycated albumin were measured using an automated analyzer, HLC-723^®^ G11 (TOSHO CO., LTD., Tokyo, Japan) [21]. Serum low-density lipoprotein-cholesterol (LDL-C) and high-density lipoprotein-cholesterol (HDL-C) levels were measured using enzymatic methods (QUALIGENT^®^ HDL-C and QUALIGENT^®^ LDL-C, SEKISUI MEDICAL. CO., LTD., Tokyo, Japan) [21]. Serum triglyceride (TG) levels were measured using enzymatic assays (Kyowa Medex, Co., Ltd., Tokyo, Japan) [21]. Free fatty acids (FFAs) were measured by a commercially available kit (Wako Chemicals, Neuss, Germany). Plasma glucose was measured in duplicate immediately after sampling on a YSI 2300 Stat Plus (YSI, Inc., Yellow Springs, OH, USA) [21]. Insulin was analyzed using a time-resolved immunofluorometric assay (AutoDELFIA Insulin kit, catalog no. B080–101, PerkinElmer, Turku, Finland) [21]. Homeostasis model assessment–insulin resistance (HOMA-IR) was calculated by the formula—fasting serum insulin (μU/mL)/fasting plasma glucose (mg/dL)/405. Serum interleukin-6 (IL-6) was measured by Human IL-6 CLEIA (Chemiluminescent Enzyme Immuno Assay) Fujirebio (Tokyo, Japan) and high-sensitivity C-reactive protein (hsCRP) was measured by a nephelometry method using N-Latex CRPII (Siemens Healthineers, Tokyo, Japan) [21].

### 2.7. Sirt1 mRNA Expression in Isolated Peripheral Blood Mononuclear Cells (PBMNCs)

PBMNCs were collected from 20 mL of heparinized blood at the beginning and after 8 weeks of the study and isolated using Histopaque-1077 (Sigma-Aldrich, St. Louis, MO, USA), as previously described [22]. PBMNCs were washed three times with phosphate-buffered saline (PBS) (−) and suspended in TRIzol reagent (Thermo Fisher Scientific, Waltham, MA, USA) for quantitative real-time PCR. Total RNA was isolated from isolated PBMNCs, cDNA synthesis and quantitative real-time PCR were performed as previously described [22]. TaqMan probes for Sirt1 were purchased from Thermo Fisher Scientific (Waltham, MA, USA). The analytical data were adjusted to the level of 18S mRNA expression as an internal control.

### 2.8. THP-1 Cell Culture

Human monocytes (THP-1 cells) obtained from ATCC were cultured in RPMI medium with 10% fetal calf serum [23]. After 16 h of serum starvation, THP-1 cells were treated with RWE 166, 332 and 3320 ng/mL (including 68, 136 or 1360 ng/mL polyphenols and 9.6, 19.2 and 192 ng/mL resveratrol, respectively) or Dimethyl sulfoxide (DEMSO) as a control for 24 h. Western blotting was performed using antibodies against Sirt1 (1:1000), phosphor(p)-AMPKα (Thr 172) (1:1000), AMPKα (1:1000) and β-actin (1:1000), as previously described [23]. The anti-rabbit polyclonal p62 antibody (PM045) was obtained from Medical & Biological Laboratories (Nagoya, Japan). Anti-phospho (p)-AMPKα (Thr 172), AMPKα and β-actin antibodies were obtained from Cell Signaling Technology Inc. (Danvers, MA, USA) and anti-Sirt1 antibodies were obtained from Millipore (Bedford, MA, USA).

### 2.9. Statistical Analysis

Data are presented as the means ± the standard deviation (SD) unless otherwise indicated. The results obtained at baseline and after 8 weeks of RWE supplementation, as well as changes within a group, were compared using a paired *t*-test. One-way ANOVA followed by Tukey’s multiple comparison test was used to determine the significance of pairwise differences among three or more groups. The correlation of two variables was analyzed by a single linear regression analysis as a Pearson correlation coefficient. Statistical significance was defined as *p* < 0.05 and statistical analyses were performed using StatMate5.

## 3. Results

### 3.1. Characteristics at Baseline and after Supplementation with RWE for 8 Weeks

The physical characteristics of the participants are shown in Table 1. BW and body mass index (BMI) were not significantly different between the baseline and the end of supplementation with RWE. Body composition, including fat mass, %fat and skeletal fat, showed no significant change between baseline and at 8 weeks of RWE supplementation. Systolic and diastolic BP and HR also showed no change between baseline and after supplementation with RWE. Fasting plasma glucose and serum insulin levels showed no significant change between baseline and the end of RWE supplementation (Table 2). However, HOMA-IR was significantly decreased after supplementation with RWE compared to the baseline. Additionally, the levels of serum TG and LDL-C were significantly decreased and serum HDL-C and FFA levels showed no differences after supplementation with RWE compared to those at baseline. Among the inflammatory markers, serum hsCRP levels were not changed; however, serum IL-6 levels showed significant decreases after 8 weeks of supplementation with RWE from baseline (Table 2). Liver function tests, such as AST, ALT and γ-GTP and kidney function tests, such as serum creatinine and uric acid, exhibited no significant change between baseline and the end of supplementation with RWE.

### 3.2. Change in Sirt1 Expression in Isolated PBMNCs after Supplementation with RWE and the Relationship with the Change in HOMA-IR

Supplementation with RWE for 8 weeks significantly increased Sirt1 mRNA expression in isolated PBMNCs compared to baseline (Figure 1A). Additionally, the relationship between the rate of change in Sirt1 expression in isolated PBMNCs (Δ%Sirt1 mRNA expression) and the rate of change in HOMA-IR (Δ%HOMA-IR) from Pearson’s correlation coefficient analysis showed a positive correlation (r = 0.6518, *p* = 0.0216) between baseline and the end of RWE supplementation (Figure 1B).

### 3.3. RWE Increased Sirt1 and p-AMPK Expression in Cultured THP-1 Cells

We evaluated whether RWE induced Sirt1 and p-AMPK expression in cultured human THP-1 cells. The administration of RWE at 166, 332 and 3320 ng/mL (including 68, 136 or 1360 ng/mL polyphenols and 9.6, 19.2 and 192 ng/mL resveratrol) in cultured THP-1 cells for 24 h significantly increased both Sirt1 and p-AMPK expression (Figure 2A–C). In addition, we confirmed that those RWE concentrations were non-toxic to cultured THP-1 cells by (data not shown).

## 4. Discussion

In this study, we demonstrated that supplementation with RWE for 8 weeks significantly increased insulin sensitivity, which was evaluated by HOMA-IR in humans. Additionally, after supplementation with RWE, serum IL-6 concentration was significantly reduced and showed a decrease in the levels of LDL-C and TG. Moreover, RWE supplementation enhanced Sirt1 expression in isolated PBMNCs, which was associated with an increase in insulin sensitivity.

Previous clinical evidence suggests that red wine consumption exerts beneficial effects on glucose metabolism, including insulin sensitivity. Da Luz et al. showed that regular red wine drinkers (at least one glass of red wine 4–5 days/week for 5 years) have lower glucose levels and a lower occurrence of diabetes than abstainers [24]. Additionally, Napoli et al. demonstrated that red wine consumption (360 mL/day) for 2 weeks markedly improved insulin resistance in patients with T2DM compared to the control group [25]. Chiva-Blanch et al. also compared the effect of moderate consumption of red wine (30 g alcohol/day), dealcoholized red wine and gin on glucose metabolism in 67 men with high cardiovascular disease for 4 weeks [26]. Red wine and dealcoholized red wine but not gin exhibited decreases in plasma insulin levels and HOMA-IR [26]. In this study, we also demonstrated that the values of HOMA-IR were significantly reduced after the administration of RWE containing 136 mg polyphenols per day for 8 weeks in nondiabetic humans.

Since red wine is rich in polyphenolic compounds, including flavonoids (anthocyanins, tannins and catechin) and nonflavonoids (stilbenes such as resveratrol, tyrosol and hydroxytyrosol) [9], the beneficial effects of red wine are thought to be exerted through polyphenols. Among the polyphenols, resveratrol has been one of the most extensively studied as a critical constituent that contributes to the health benefits of red wine. Previous studies demonstrated that resveratrol might play potential therapeutic roles in cardiometabolic health through multiple mechanisms, such as anti-inflammatory, antioxidant and anti-diabetic effects, which are mediated by the activation of Sirt1, estrogen receptor (ER) signaling, nuclear factor-erythroid-derived 2-related factor-2 (Nrf2) or AMPK [20,27,28,29]. Several reports showed that catechin, epicatechin, quercetin and anthocyanin also can activate Sirt1 or AMPK [30,31,32,33], however, the number of reports is so few, compared to those of resveratrol. In this study, we focused on Sirt1, which is an important regulator of a wide variety of cellular processes, including glucose/lipid metabolism and anti-inflammation, via the deacetylation of many substrates [34,35]. Our data showed the increased expression of Sirt1 in PBMNCs after supplementation with RWE including 19.2 mg resveratrol. Additionally, the levels of Sirt1 expression in PBMNCs had a positive relationship with insulin sensitivity, which was evaluated by HOMA-IR. Additionally, serum IL-6 was reduced after the administration of RWE including resveratrol. A previous report showed that decreased Sirt1 expression levels in circulating monocytes are correlated with insulin resistance in humans [36]. Moreover, Gillum et al. reported that Sirt1 expression was reduced in adipose tissues of obese males with insulin resistance and mRNA expression of CD14, a macrophage marker, in adipose tissue is negatively correlated with Sirt1 expression [37]. Chronic low-grade tissue inflammation is an important etiological component of insulin resistance [38]. Elevated levels of proinflammatory cytokines, such as IL-6, in the blood have been detected in individuals with insulin resistance. The activation of monocytes/macrophages in the circulation and adipose tissue has been demonstrated to lead to the release of various inflammatory mediators. Sirt1 may contribute to the negative regulation of inflammation in several tissues or cells, including monocytes/macrophages, through the deacetylation of NF-κB (p65 subunit) [39,40,41]. Therefore, the effect of RWE including resveratrol on insulin resistance and inflammation may be associated with increased Sirt1 expression in PBMNCs. However, we could not show the levels of acetylated NF-κB (p65) in mononuclear cells or the relationship between Sirt1 expression and serum IL-6 values.

In addition to reducing inflammation, previous reports showed that Sirt1 may positively regulate insulin signaling by interacting with tyrosine phosphatase 1B, insulin receptor substrate or phosphoinositide 3-kinase in insulin-sensitive tissues such as skeletal muscle [42,43,44]. Timmers et al. also reported that resveratrol supplementation (150 mg/day) for 30 days in obese humans increased insulin sensitivity, improved muscle mitochondrial respiration and activated Sirt1 and AMPK in skeletal muscle [45]. Additionally, Liu et al. reported that resveratrol inhibited inflammation and ameliorated insulin-resistant endothelial dysfunction through AMPK and Sirt1 [46]. In this study, we demonstrated that the administration of RWE including resveratrol increased the expression of Sirt1 and p-AMPK in cultured THP-1 cells. However, we could not evaluate Sirt1 expression levels or inflammation in other tissues/cells, such as skeletal muscle, adipose tissue or endothelial cells.

On the other hand, other reports indicate that resveratrol has no effects on insulin sensitivity. Yoshino et al. demonstrated that oral resveratrol (75 mg/day) supplementation in nonobese and postmenopausal women with normal glucose tolerance did not improve metabolic function, including insulin sensitivity [47]. Moreover, Poulsen et al. reported that resveratrol (500 mg/day) supplementation in obese men had no effect on insulin sensitivity [48]. Thus, the efficacy of resveratrol for insulin sensitivity is controversial in humans. Therefore, the beneficial effects of RWE may be attributed to the overall mix of all of its components and not to a specific action of one, such as resveratrol.

Sirt1 regulates lipid metabolism through the modulation of sterol regulatory element-binding protein (SREBP)-1C activity, liver X-receptor (LXR) and farnesoid X receptor (FXR) via deacetylation of those molecules [49,50,51]. Therefore, in this study, RWE including resveratrol might contribute to deceased levels of LDL-C and TG through Sirt1 activation, in addition to increased insulin sensitivity.

There are several limitations in this study. First, this study is a single-arm, open-label study with small sample size and occurred over a short time period. Second, participants in this study are individuals who are interested in health and the supplements. Therefore, the bias on the results in this study cannot be eliminated. However, the results from in vitro experiments showing that RWE increases Sirt1 expression can support the results in PBMNCs of this study. Third, we could not measure the concentration of polyphenols, including resveratrol, in circulation. Fourth, we evaluated Sirt1 expression in only PBMNCs, not in other tissues/cells. Lastly, we evaluated insulin resistance only by the calculation of HOMA-IR, although the gold standard method for assessment of insulin sensitivity is a hyperinsulinemic-euglycemic clamp study.

## 5. Conclusions

Supplementation with RWE improved metabolism, such as insulin sensitivity, lipid profile and inflammation, in nondiabetic humans. Additionally, RWE supplementation induced an increase in Sirt1 expression in PBMNCs, which may be associated with an improvement in insulin sensitivity. However, further study including a randomized control trial or a cross over trial will be required to confirm these results.

## Figures and Tables

**Figure 1 nutrients-12-03108-f001:**
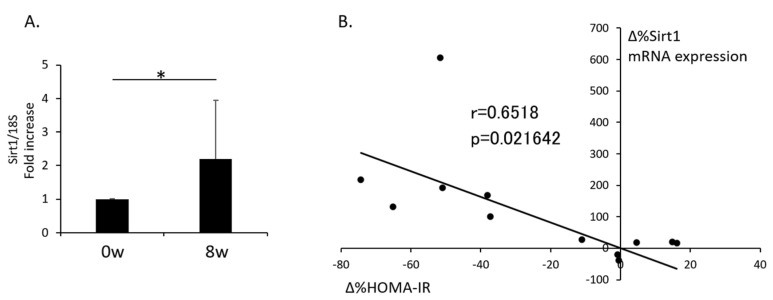
Change in mRNA expression of Sirt1 in PBMNCs after supplementation with red wine extract and the relationship between the change in Sirt1 expression and insulin sensitivity. (**A**) mRNA expression of Sirt1 normalized to 18S levels in isolated PBMNCs (*n* = 12). The data shown are the means ± the standard deviations. * *p* < 0.05 vs. the indicated groups. (**B**) The relationship between the rate of change in Sirt1 expression in isolated PBMNCs (Δ%Sirt1 mRNA expression) and the rate of change in HOMA-IR (Δ%HOMA-IR) from Pearson’s correlation coefficient analysis (*n* = 12). PBMNCs: peripheral blood mononuclear cells, HOMA-IR: homeostasis model assessment–insulin resistance.

**Figure 2 nutrients-12-03108-f002:**
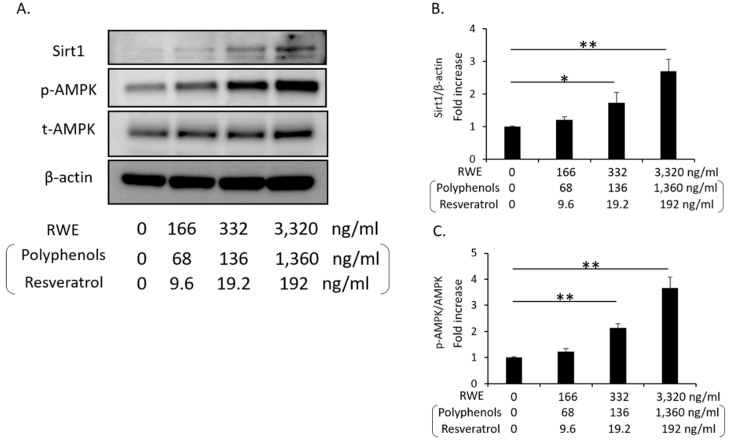
Change in Sirt1 and p-AMPK expression by the administration of red wine extract in cultured THP-1 cells. (**A**) Representative western blots of Sirt1, p-AMPK, AMPK and β-actin in cultured THP-1 cells (*n* = 4). (**B**) Quantitative ratios of Sirt1 to β-actin (*n* = 4). (**C**) Quantitative ratios of p-AMPK to AMPK (*n* = 4). The data shown are the means ± the standard deviations. * *p* < 0.05, ** *p* < 0.01 vs. the indicated groups. AMPK: AMP-activated kinase, RWE: red wine extract.

**Table 1 nutrients-12-03108-t001:** Characteristics of participants at baseline and after supplementation with red wine extract.

	0 Week	8 Weeks	*p* Value
Age	47.5 ± 11.3		
Male:female	8:4		
Body weight (kg)	66.6 ± 16.4	66.0 ± 16.1	0.282
Body mass index (kg/m^2^)	23.3 ± 3.8	23.1 ± 3.7	0.278
Systolic blood pressure (mmHg)	118.8 ± 13.5	119.7 ± 13.3	0.658
Diastolic blood pressure (mmHg)	71.5 ± 10.4	73.3 ± 13.0	0.580
Heart rate (/min)	71 ± 8.7	71.4 ± 5.2	0.860
Fat mass (kg)	18.8 ± 8.8	17.8 ± 8.4	0.067
%fat	25.7 ± 9.3	24.1 ± 7.0	0.255
Skeletal muscle mass (kg)	26.1 ± 6.4	26.5 ± 6.8	0.369

**Table 2 nutrients-12-03108-t002:** Laboratory data of participants at baseline and after supplementation with red wine extract.

	0 Week	8 Weeks	*p* Value
Fasting plasma glucose (mg/dL)	95.6 ± 8.2	89.6 ± 9.3	0.100
Fasting serum insulin (μU/mL)	7.06 ± 5.49	4.88 ± 3.88	0.063
HOMA-IR	1.71 ± 1.38	1.13 ± 1.03	0.046
HbA1c (%)	5.2 ± 0.5	5.2 ± 0.4	0.135
Glycated albumin (%)	13.8 ± 1.2	13.8 ± 1.1	0.431
LDL-C (mg/dL)	119.7 ± 21.0	114.7 ± 19.6	0.013
HDL-C (mg/dL)	56.0 ± 19.0	58.0 ± 11.3	0.097
TG (mg/dL)	246.9 ± 285.6	182.2 ± 220.5	0.032
log-TG	2.12 ± 0.48	2.02 ± 0.40	0.034
Free fatty acid	515.0 ± 309.8	549.5 ± 177.6	0.688
log free fatty acid	2.62 ± 0.30	2.72 ± 0.16	0.305
IL-6 (ng/mL)	1.8 ± 0.8	1.4 ± 0.6	0.019
hsCRP (mg/dL)	1049.7 ± 1620.2	1158.3 ± 1638.6	0.400
log-hsCRP	2.70 ± 0.52	2.73 ± 0.58	0.666
AST (IU/mL)	21.3 ± 7.3	21.3 ± 7.4	0.352
ALT (IU/mL)	17.0 ± 18.7	20.0 ± 20.8	1.000
γ-GTP (IU/mL)	22.3 ± 71.1	24.0 ± 59.2	0.435
Cr (mg/dL)	0.72 ± 0.16	0.69 ± 0.15	0.054
Uric acid (mg/dL)	6.1 ± 2.2	6.1 ± 2.0	0.574

HOMA-IR: homeostasis model assessment–insulin resistance, LDL-C: low-density lipoprotein-cholesterol, HDL-C: high-density lipoprotein-cholesterol, TG: triglyceride, IL-6: interleukin-6, hsCRP: high-sensitivity C-reactive protein, AST: aspartate aminotransferase, ALT: alanine transaminase, γ-GTP: γ-glutamyl transpeptidase, Cr: creatinine, UA: uric acid.

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
