# Peer review of "Supplementation with Red Wine Extract Increases Insulin Sensitivity and Peripheral Blood Mononuclear Sirt1 Expression in Nondiabetic Humans"

_nutrients, 2020, doi:10.3390/nu12103108_

Round 1

Reviewer 1 Report

This article elucidates the role of red wine extract (RWE) supplementation on insulin sensitivity in nondiabetic humans. The authors also aim to understand the possible mechanisms by which Sirt1 expression is influenced by the RWE. The authors have used both in vivo and in vitro methods to undertake the study. The authors have given a good overview of the project in the abstract. Overall the manuscript is written well but will need few more details. The materials and methods section is missing a vital information on the RWE used in this study.

Major Comment

  1. This article is primarily based on the red wine extract (RWE) obtained from a commercial company in Japan. The details of the extract and its polyphenolic profile is a very critical information that would provide a thorough knowledge on the compound being investigated and strengthen the manuscript. There is only a limited information provided by the authors in lines # 68-69 under section 2.1 on the composition of RWE.

The authors should either add more information on the polyphenolic profile of the RWE besides stating the amount of resveratrol and polyphenols. If the RWE used in this study has been previously characterised, the authors should provide a reference for this, as this is a very crucial information to improve the article.

  1. Cell culture experiments using THP-1 cells is missing a vital information on the cytotoxicity of RWE on these THP-1 cells. In an in vitro model, it is essential to do a cytotoxicity of the compound (RWE) being tested prior to analysis to rule out any confounding factors that maybe influencing the results observed.

So, it is expected that authors either include this data/ information in the revised version or provide a reference to clearly state that the RWE concentrations used in this study is non-cytotoxic to THP-1 cells.

Minor comments

  1. Lines 48-52 on CR/DR mimetics needs a bit of explanation as to how this information is connected to the current study. If there is no relevance to this study, then it is recommended to reduce the description on CR/DR mimetics.
  2. The use of phrase “red wine extract including resveratrol” has been stated at multiple times throughout the manuscript (For example in line#63). This phrase needs to be restructured as the RWE used contains other polyphenols too and over emphasis on resveratrol is not required.

    3.The last sentence of Introduction line #61-63 needs to be restructured as       “including resveratrol” is been used twice in the same sentence.

  1. In section 2.2, line# 79, it is recommended that the authors clearly state how were the participants instructed to abstain from supplements (Were they given a diary to keep the log etc.)
  2. In section 2.8 describing THP-1 cell culture, line 133 says DEMSO. It should say DMSO and if DMSO has not been elaborated in the article, it should be done.
  3. In the results section the tables have been referred to in the text in 3.1 section. The authors should do the same for section 3.2 and 3.3 where the figures have not been referred to in the texts of 3.2 and 3.3
  4. The discussion section needs to discuss about the synergistic effects of various polyphenols including resveratrol that are found in this RWE and are responsible for observed effect.
  5. The limitations in line from 261-269 needs to be paraphrased. It is recommended that the authors say that limitations in this study as a descriptive paragraph and not just list them in numbers. Authors should also state that how this article is relevant despite of all these limitations.

Author Response

Response to reviewers

----- Here we sincerely appreciated editor’s suggestions of our manuscript. We did response all the concerns from all reviewers. Reviewers’ comments were helpful and also making our manuscript quality better one.

Reviewer 1

This article elucidates the role of red wine extract (RWE) supplementation on insulin sensitivity in nondiabetic humans. The authors also aim to understand the possible mechanisms by which Sirt1 expression is influenced by the RWE. The authors have used both in vivo and in vitro methods to undertake the study. The authors have given a good overview of the project in the abstract. Overall the manuscript is written well but will need few more details. The materials and methods section is missing a vital information on the RWE used in this study.

Major Comment

  1. This article is primarily based on the red wine extract (RWE) obtained from a commercial company in Japan. The details of the extract and its polyphenolic profile is a very critical information that would provide a thorough knowledge on the compound being investigated and strengthen the manuscript. There is only a limited information provided by the authors in lines # 68-69 under section 2.1 on the composition of RWE. The authors should either add more information on the polyphenolic profile of the RWE besides stating the amount of resveratrol and polyphenols. If the RWE used in this study has been previously characterised, the authors should provide a reference for this, as this is a very crucial information to improve the article.

Ans)
We appreciate for reviewer’s constructive comments on our manuscript. We inserted the detailed information of RWE, as below.
“The polyphenol contents in the RWE were assessed by high-performance liquid chromatography (HPLC) analysis (Supplemental Figure) and revealed that 166 mg of wine solids contained 9.6 mg of resveratrol and 68 mg of polyphenols; catechin 1.16 mg, epicatechin 0.83 mg, tannin 29.3 mg, quercetin glycoside 0.33 mg, malvidin glycoside 1.99 mg, total anthocyanin 5.15 mg, anthocyanin monomer 4.15 mg, anthocyanin polymer 1.00 mg per 1 capsule.”

  1. Cell culture experiments using THP-1 cells is missing a vital information on the cytotoxicity of RWE on these THP-1 cells. In an in vitro model, it is essential to do a cytotoxicity of the compound (RWE) being tested prior to analysis to rule out any confounding factors that maybe influencing the results observed. So, it is expected that authors either include this data/ information in the revised version or provide a reference to clearly state that the RWE concentrations used in this study is non-cytotoxic to THP-1 cells.

Ans)

We appreciate for reviewer’s constructive comments on our manuscript. We confirmed that the RWE concentration used in this study were non-cytotoxic to cell cultured THP-1 cells, as below.

Minor comments

  1. Lines 48-52 on CR/DR mimetics needs a bit of explanation as to how this information is connected to the current study. If there is no relevance to this study, then it is recommended to reduce the description on CR/DR mimetics.

Ans)

Resveratrol is one of the CR/DR mimetics because resveratrol can positively  modulate lifespan extension and cardiometabolic health through induction of Sirt1 activation. Therefore, red wine extract including resveratrol can be also CR/DR mimetics, as Sirt1 activator. We explained these points in the section of Introduction.

We added “as one of the CR/DR mimetics”, as below.

“Resveratrol, a polyphenolic phytoalexin that occurs in red wine, has been one of the most extensively studied Sirt1 activators, as one of the CR/DR mimetics [20] and is a critical constituent that contributes to the health benefits of red wine.”

  1. The use of phrase “red wine extract including resveratrol” has been stated at multiple times throughout the manuscript (For example in line#63). This phrase needs to be restructured as the RWE used contains other polyphenols too and over emphasis on resveratrol is not required.

Ans)

We changed “red wine extract including resveratrol” to “red wine polyphenols” or “red wine extract”.

  1. The last sentence of Introduction line #61-63 needs to be restructured as  “including resveratrol” is been used twice in the same sentence.

Ans)
We changed “red wine extract including resveratrol” to “red wine polyphenols” or “red wine extract”.

  1. In section 2.2, line# 79, it is recommended that the authors clearly state how were thelog etc.)
    Ans) 
  2. During the study period, participants were instructed to abstain from supplements and foods suspected to contain polyphenols in significant amounts, and the adherence for them was confirmed every visit.
  3. participants instructed to abstain from supplements (Were they given a diary to keep the
  4. In section 2.8 describing THP-1 cell culture, line 133 says DEMSO. It should say DMSO and if DMSO has not been elaborated in the article, it should be done. 
  5. Ans)
    We inserted “Dimethyl sulfoxide (DEMSO)”.
  6. In the results section the tables have been referred to in the text in 3.1 section. The autreferred to in the texts of 3.2 and 3.3
    Ans) 
  7. We inserted “Figure 1A” and “Figure 1B”, as reference.
  8. hors should do the same for section 3.2 and 3.3 where the figures have not been
  9. The discussion section needs to discuss about the synergistic effects of various polyphenols including resveratrol that are found in this RWE and are responsible for observed effect.We inserted the sentence, as below.
  10. Several reports showed that catechin, epicatechin, quercetin and anthocyanin also can activate Sirt1 or AMPK [31-34], however, the number of reports is so few, compared to those of resveratrol.
  11. Ans)
  • [31] Zhao, S.; Zhang, L.; Yang, C.; Li, Z.; Rong. S. Procyanidins and Alzheimer's Disease. Mol Neurobiol 2019, 56, 5556-5567.
  • [32 ] Leyton, L.; Hott, M.; Acuña, F.; Caroca, J.; Nuñez, M.; Martin, C.; Zambrano, A.; Concha, M.I.; Otth, C.
  • Nutraceutical activators of AMPK/Sirt1 axis inhibit viral production and protect neurons from neurodegenerative events triggered during HSV-1 infection. Virus Res 2015, 205, 63-72.
  • [33] Sayed, A.M.; Hassanein, E.H.M.; Salem, S.H.; Hussein, O.E.; Mahmoud, A.M. Flavonoids-mediated SIRT1 signaling activation in hepatic disorders. Life Sci 2020, 259, 118173.
  • [34] de Sousa Moraes, L.F.; Sun, X.; Peluzio, M.D.C.G.; Zhu, M.J. Anthocyanins/anthocyanidins and colorectal cancer: What is behind the scenes? Crit Rev Food Sci Nutr 2019, 59, 59-71.
  1. The limitations in line from 261-269 needs to be paraphrased. It is recommended that the authors say that limitations in this study as a descriptive paragraph and not just list them in numbers. Authors should also state that how this article is relevant despite of all these limitations.

Ans)
We changed to the sentences regarding the limitations of this study, as below.
“There are several limitations in this study. First, this study is a single-arm, open-label study with small sample size and occurred over a short time period. Second, participants in this study are individuals who are interested in health and the supplements. Therefore, the bias on the results in this study cannot be eliminated. However, the results from in vitro experiments showing that RWE increases Sirt1 expression can support the results in PBMNCs of this study. Third, we could not measure the concentration of polyphenols, including resveratrol, in circulation. Fourth, we evaluated Sirt1 expression in only PBMNCs, not in other tissues/cells. Lastly, we evaluated insulin resistance only by the calculation of HOMA-IR.”
In addition, we added “However, further study including a randomized control trial, or a cross over trial will be required to confirm these results.”, in the section of Conclusions.

Reviewer 2 Report

The authors describe the potential benefits on glucose/lipid metabolism from short-term administration of red wine extracts in healthy humans over an 8-week period. The authors find that the HOMA-IR significantly improved, along with improved levels of LDLc and triglycerides. The authors also find significantly reduced levels of IL-6 as well as increased Sirt-1 mRNA expression in the PBMNCs. The authors also show, through in vitro experiments, that red wine extract treatment increased Sirt-1 and pAMPK expression in THP-1 cells. The paper overall is clearly written, the methods are overall detailed, and the discussion of their findings are supported by the results of this study. The following aspects need to be addressed.

Introduction:

In line 40, it would be useful to quantify what constitutes an ‘appropriate consumption’ of red wine.

Methods:

Under ‘Subjects and study design’, it would be useful for the authors to mention in which institute the study was conducted.

The use of the term ‘obvious’ in the selection criteria (such as obvious endocrine disease, or obvious anemia) is a bit non-descriptive. In line 72, the term ‘obvious’ can be replaced with ‘pre-existing’. In line 73, for ‘obvious anemia’, it would be better to define what was the cut-off for defining anemia.

For the measurement of HOMA-IR, it would be useful to provide the actual formula rather than mentioning that the ‘standard formula’ was used.

It would be useful to mention under section 2.7 at what time points the PBMNCs were collected from the study population.

In section 2.8, please provide the expansion for DEMSO.

Under the ‘statistical analysis’ section, for the one-way repeated measures ANOVA, it would be useful to clarify what exactly ‘the three groups’ were. Rather, the word ‘the’ can be deleted, and the term can just be changed to ‘three or more groups’

Results:

Given that IL-6 production and Sirt1 regulation are some of the key components of an inflammatory state, the authors could perform a correlation analysis between % changes in IL-6 and Sirt-1 mRNA expression.

In line 190, the authors mention ‘monocytes and THP-1 cells’. This is confusing. It is better to delete the word ‘monocytes’.

Discussion:

The authors have mentioned the quantity of resveratrol and other polyphenols in red wine extract they procured for this experiment in the methods section. But what is the quantity of the same polyphenols in a typical glass of red wine? In other words, was the quantity of resveratrol/polyphenols much higher (or lower) in this study’s red wine extract compared to the, say, a glass of red wine? This would be an interesting discussion as well.

Have other studies on resveratrol or red wine polyphenols shown any effects on hs-CRP levels?

For the fifth point of limitation, the authors could expand on it mentioning that methods such as frequently sampled intravenous glucose tolerance tests or hyperinsulinemic-euglycemic clamp studies were not utilized to assess insulin sensitivity.

Author Response

Response to reviewers

----- Here we sincerely appreciated editor’s suggestions of our manuscript. We did response all the concerns from all reviewers. Reviewers’ comments were helpful and also making our manuscript quality better one.

Reviewer 2

The authors describe the potential benefits on glucose/lipid metabolism from short-term administration of red wine extracts in healthy humans over an 8-week period. The authors find that the HOMA-IR significantly improved, along with improved levels of LDLc and triglycerides. The authors also find significantly reduced levels of IL-6 as well as increased Sirt-1 mRNA expression in the PBMNCs. The authors also show, through in vitro experiments, that red wine extract treatment increased Sirt-1 and pAMPK expression in THP-1 cells. The paper overall is clearly written, the methods are overall detailed, and the discussion of their findings are supported by the results of this study. The following aspects need to be addressed.

Introduction:

In line 40, it would be useful to quantify what constitutes an ‘appropriate consumption’ of red wine.
Ans)
We inserted “20-30g/day as amount of alcohol”, as below.
“Appropriate consumption of red wine, 20-30g/day as amount of alcohol, is thought to be part of a healthy lifestyle [2-4].”

Methods:

Under ‘Subjects and study design’, it would be useful for the authors to mention in which institute the study was conducted.
Ans)
We inserted “and conducted at Kanazawa Medical University Hospital.”, as below.
This study is a single-arm, open-label, prospective study, and conducted at Kanazawa Medical University Hospital.

The use of the term ‘obvious’ in the selection criteria (such as obvious endocrine disease, or obvious anemia) is a bit non-descriptive. In line 72, the term ‘obvious’ can be replaced with ‘pre-existing’. In line 73, for ‘obvious anemia’, it would be better to define what was the cut-off for defining anemia.
Ans)
We changed “obvious” to “preexisting”, and we showed the cut-off for defining anemia.

For the measurement of HOMA-IR, it would be useful to provide the actual formula rather than mentioning that the ‘standard formula’ was used.
Ans)
We changed to “Homeostasis model assessment–insulin resistance (HOMA-IR) was calculated by the formula: fasting serum insulin (μU/ml) / fasting plasma glucose (mg/dl)/405.”.

It would be useful to mention under section 2.7 at what time points the PBMNCs were collected from the study population.
Ans)
We added “at the beginning and after 8 weeks of the study”, as below.
PBMNCs were collected from 20 ml of heparinized blood at the beginning and after 8 weeks of the study and isolated using Histopaque-1077 (Sigma-Aldrich, St. Louis, MO, USA), as previously described.

In section 2.8, please provide the expansion for DEMSO.
Ans)
We inserted “Dimethyl sulfoxide (DEMSO)”.

Under the ‘statistical analysis’ section, for the one-way repeated measures ANOVA, it would be useful to clarify what exactly ‘the three groups’ were. Rather, the word ‘the’ can be deleted, and the term can just be changed to ‘three or more groups’
Ans)
We changed to ‘three or more groups’.

Results:

Given that IL-6 production and Sirt1 regulation are some of the key components of an inflammatory state, the authors could perform a correlation analysis between % changes in IL-6 and Sirt-1 mRNA expression.
Ans)
We showed a result of a correlation analysis between %changes in serum IL-6 and Sirt1 mRNA expression. There was not significantly relationship between them.

In line 190, the authors mention ‘monocytes and THP-1 cells’. This is confusing. It is better to delete the word ‘monocytes’.
Ans)
We deleted “monocytes”.

Discussion:

The authors have mentioned the quantity of resveratrol and other polyphenols in red wine extract they procured for this experiment in the methods section. But what is the quantity of the same polyphenols in a typical glass of red wine? In other words, was the quantity of resveratrol/polyphenols much higher (or lower) in this study’s red wine extract compared to the, say, a glass of red wine? This would be an interesting discussion as well.
Ans)

Have other studies on resveratrol or red wine polyphenols shown any effects on hs-CRP levels?
Ans)
Several studies showed the data of hs-CRP as the assessment for systemic inflammation.
Since high-sensitivity CRP is sensitive and increases even with very mild inflammation, it is considered that two or more measurements are necessary to evaluate the change of systemic inflammation. In this study, we evaluated the levels of serum IL-6. Previously, we reported that 25% calorie restriction in human decreased in serum IL-6 levels from the baseline (Kitada et al. BBA. 2013).

For the fifth point of limitation, the authors could expand on it mentioning that methods such as frequently sampled intravenous glucose tolerance tests or hyperinsulinemic-euglycemic clamp studies were not utilized to assess insulin sensitivity.
Ans)
We inserted the sentence “Lastly, we evaluated insulin resistance only by the calculation of HOMA-IR, although the gold standard method for assessment of insulin sensitivity is a hyperinsulinemic-euglycemic clamp study.” In the section of limitations.

Reviewer 3 Report

The authors are applauded for taking on this study evaluating the metabolic benefits of red wine extract on markers of lipid and glucose metabolism. This is an interesting area of study with important implications for the use of supplements to improve health. The experiments performed provide good evidence for RWE efficacy, and the manuscript is well written. While the results of this study are promising, there are critical experimental design flaws that significantly reduce this reviewer’s enthusiasm for this manuscript. Specifically, the implementation of an open label, single arm study design, without a placebo group in otherwise healthy adults is not a sufficiently rigorous study design to avoid bias.

Major Concerns: The experimental design implemented in this study does not include a placebo group, randomization, or a cross over analysis. Given the experimental population and the nature of the supplement, this is a particularly weak design which cannot eliminate bias.

Author Response

Response to reviewers

----- Here we sincerely appreciated editor’s suggestions of our manuscript. We did response all the concerns from all reviewers. Reviewers’ comments were helpful and also making our manuscript quality better one.

Reviewer 3

The authors are applauded for taking on this study evaluating the metabolic benefits of red wine extract on markers of lipid and glucose metabolism. This is an interesting area of study with important implications for the use of supplements to improve health. The experiments performed provide good evidence for RWE efficacy, and the manuscript is well written. While the results of this study are promising, there are critical experimental design flaws that significantly reduce this reviewer’s enthusiasm for this manuscript. Specifically, the implementation of an open label, single arm study design, without a placebo group in otherwise healthy adults is not a sufficiently rigorous study design to avoid bias.

Major Concerns: The experimental design implemented in this study does not include a placebo group, randomization, or a cross over analysis. Given the experimental population and the nature of the supplement, this is a particularly weak design which cannot eliminate bias.
Ans)
We appreciate for the reviewer’s comment.
As the reviewer pointed out, the design of this study is weak, because of the single arm study. This point is one of the limitations of this study, as described in the section of discussion. We changed to the sentences regarding the limitations of this study, as below.
“There are several limitations in this study. First, this study is a single-arm, open-label study with small sample size and occurred over a short time period. Second, participants in this study are individuals who are interested in health and the supplements. Therefore, the bias on the results in this study cannot be eliminated. However, the results from in vitro experiments showing that RWE increases Sirt1 expression can support the results in PBMNCs of this study. Third, we could not measure the concentration of polyphenols, including resveratrol, in circulation. Fourth, we evaluated Sirt1 expression in only PBMNCs, not in other tissues/cells. Lastly, we evaluated insulin resistance only by the calculation of HOMA-IR, although the gold standard method for assessment of insulin sensitivity is a hyperinsulinemic-euglycemic clamp study.”
In addition, we added “However, further study including a randomized control trial, or a cross over trial will be required to confirm these results.”, in the section of Conclusions.

Round 2

Reviewer 1 Report

The authors have now addressed all of my comments in this revised version.

Reviewer 3 Report

The inclusion of additional limitations to the current study design are made, with the addition of a statement specifically noting the necessity of placebo controlled clinical trials. This is important, as the experimental design (single arm, no placebo) are the most significant weaknesses of this study.